# Historical Transition of a Farming System towards Industrialization: A Danish Agricultural Case Study Comparing Sustainability in the 1840s and 2019

Nele Lohrum *, Morten Graversgaard  and Chris Kjeldsen

Department of Agroecology, Aarhus University, Blichers Allé 20, 8830 Tjele, Denmark;
morten.graversgaard@agro.au.dk (M.G.); chris.kjeldsen@agro.au.dk (C.K.)
* Correspondence: nele.lohrum@agro.au.dk; Tel.: +45-41-65-55-70

**Abstract:** A Danish pre-industrial farming system is reconstructed and compared to its modern industrialized farming system equivalent to evaluate agricultural performance in a sustainability perspective. The investigated Danish farm system and its contributing elements have undergone significant transformations. The intensity of contemporary agriculture shows that high productivity levels have been achieved by increasing the input of energy using modern machinery. At the same time, the energy efficiency (calculations based on energetic indicators) diminishes over time as the degree of dependence on fossil fuels increases. The results from this study show significant changes in the farming system, specifically inputs from agricultural land use, livestock, and energy systems. From being highly circular, the system changed to being a clear linear farming system with highly increased productivity but less efficient at the same time, questioning the relationship between productivity and efficiency and resource utilization in modern farming systems. Through utilizing an agroecological historical approach by comparing system performance over time, the results offer opportunities to explore how agricultural farming systems evolve over time and help to describe the complexity of the system level in a sustainability perspective.

**Keywords:** agroecology; agricultural development; Danish farming system; historical agriculture; rural studies; sustainability



## 1. Introduction

At present, agricultural systems face severe environmental, economic, social, and institutional challenges [1–3]. These societal challenges are a result of human development, increasing agricultural intensification, specialization, and mechanization [4–6], with the outcome of very productive agricultural systems but also systems with severe environmental, climatic and social impacts [7–10]. Apparently, the modernization and industrialization of farming systems have radically changed the patterns of agricultural production and human consumption, as well as how we as humans perceive agri-food systems. In the scientific literature on transitions of agriculture, Goodman, Sorj [4] conclude that structural changes and agricultural development, such as appropriation and substitutionism, have contributed to a radical transformation of contemporary food systems compared to pre-industrial systems. Fonte [11] argues that through modernization the systems are distinguished by both structural as well as cultural change, where new cultural understandings shape new modes of organizing these food systems. According to Lemaire, Ryschawy, De Faccio Carvalho, Gastal [12], specialization has been a major factor in the intensification of agriculture in recent years, where production capacity is enhanced by intensifying production at specific production sites. Aspects of intensive agriculture and food production systems, however, may be considered different, depending on the perspective and purpose. As stated by Cox, Atkins [5] and Woodhouse [13], population growth and urbanization have necessitated the increased productivity of agricultural systems.

Before industrialization, peasant farming was an essential factor in population development and production [14]. Danish agriculture, particularly before industrialization, was a circular farming system with limits to the local environment [15]. Shifts in land use categories have occurred as an outcome of driving forces, such as technology and land improvement [16]. The particularly great economic success of the conversion of Danish agriculture to animal production can be found in the design and distribution of the cooperative dairy movement [17]. Since the industrial revolution, innovations such as modern machinery and inorganic fertilizers have helped to reduce the labor force, or even appropriated human labor and draft animals [6,18,19], but at the same time resulted in a dependency between human activity and fossil energy use to maintain agricultural systems [18–20]. These system changes came along with broader societal change, altogether constituting a new phase of Danish social ecology [21]. Denmark has, since the late nineteenth century, been distinguished by industrialized farming, a high share of agricultural land use, and an export-oriented agricultural sector [22–24]. Through these development phases, Danish agriculture became the modern farming agriculture of recent time. For these reasons, "reconstructing" and comparing Danish pre-industrial agriculture to industrial farming serves as an exemplary case for illustrating how the evolvement of modern farming systems creates sustainability challenges. Furthermore, it can inform sustainability assessment and intervention in contemporary farming systems.

A vast amount of studies about reconstructing past agricultural farming systems based on historical documents exists, such as maps, notebooks, statistics, and tax records [16,18,25–28]. Furthermore, there are numerous studies of changes in agro-ecosystems over time [6,29–32]. These examples of "reconstructing" and comparing past agricultural systems and landscapes with current farming systems shows that the approach of adopting historical perspectives highlights important changes in agricultural systems. Such investigations on a regional scale allows for exploring changes in spatial and socio-economic dynamics over time, while also illuminating key determinants for spatial and temporal agro-environmental change [6,27,33].

This paper seeks to contribute to extant knowledge concerning the features of sustainable farming systems [6,31] by focusing on the historical transition from a pre-industrialized to an industrialized farm system and the historical evolution of energy use by measuring (the ever increasing) dependency on fossil energy [6,30,34–36]. The Danish Kragerup Estate (KE) serves as a case study. A comparative characterization analysis of the farming system at KE was performed for two periods: the years 1837–1841 (representative for the early nineteenth century, and in the following 1840s) and 2019 (representative for a 5-year-period) [34], were investigated and assessed. The aim was to identify and describe the farming system at KE in the two periods by comparing historical data and investigating how structural elements of the specific farming system in question have evolved over time. Investigating how farming systems evolve over time is interesting from an environmental, agronomic, land use change, cultural, and historical perspective [33]. Through these approaches, this study presents a Danish case study relevant for evaluating the sustainability of farming systems over time. The results of this investigation will be discussed in the context of historical agricultural development, structural changes, linearity, and the circularity of farming systems, including a socio-ecological perspective.

*Theoretical Background*

Agroecology seeks to consider and integrate different perspectives on sustainability in order to develop integrated approaches to farming system problems [19,35]. According to Dalgaard et al. [35], agroecology is defined as: "*The study of the interactions between plants, animals, humans and the environment within agricultural systems. Agroecology as a discipline therefore covers integrative studies within agronomy, ecology, sociology and economics*". A related position is adopted by Francis, Lieblein, Gliessman et al. [36], as they emphasize that agroecology involves '*the integrative study of the ecology of the entire food systems, encompassing ecological, economic and social dimensions*'. This is further emphasized by Cattaneo,

Marull [32] in their analyzes of the social metabolism of agroecosystems. These perspectives thus encompass people, ecological conditions, energy, and materials utilization with regards to processing, food security, and the impacts of polices and regulation on market conditions in agroecosystems. In the nineteenth century, farming systems traditionally relied on peasant management with the use of local resources, remaining within its biophysical boundaries. External inputs were at a low level, or even absent, and by-products were re-used rather than wasted [6,37]. Through industrialization, many components of agroecosystems were appropriated, decreasing the human input and increasing the dependence on non-renewable energy inputs, such as fossil fuel use [6,19]. The attributes of flow of agricultural, pre-industrial bio-economy, in relationship to its surrounding environment, was distinguished by a circularity of flows. According to Cattaneo, Marull [32], a novel perspective in landscape agroecology is needed to illuminate how these shifts from a circular flowing of matter and energy towards a more linear flow are present in recent agriculture and how agroecosystems and landscape functionality might be managed sustainably in future agroecosystems. At this background, this case study approaches identifying and describing farming systems and how these are linked to a specific agroecological and social environments, which illustrates how farming systems have developed over time with regards to different perspectives of sustainability.

## 2. Materials and Methods

Kragerupgaard, today known as Kragerup Estate (KE), is located (55.51004° N, 11.37847° E) close to Ruds Vedby, Kalundborg Municipality, Southwest Zealand, Denmark. KE has a long agricultural history, which is well documented [38–41]. The KE farming system was a highly mixed farming system in the 1840s, with both crop and livestock production based on working input from peasant farming. Recently, KE is a modern farming system with a variation in crop production and livestock (pig) production. The main soil type is defined as clay till.

The 1840s and 2019 were chosen in order to compare changes in the capacity and efficiency in a farming system over time. The reconstruction of the farming system is based on historical data from the KE local archive (handwritten booklets, books, notes, and tables) which are compared and validated for methodological applicability. Land use and livestock data from 1840s, and data from KE 2019 (agricultural land use, livestock data, and energy data [42,43]), were selected for being representative of two distinct phases of the Danish farming history. Field data from 2019 are regarded as representative of industrialized, market-integrated, and high productive agriculture, largely sustained by chemical fertilizers and internationally sourced feedstuff.

Historical maps and data are useful to trace land use practices for the analysis of long-term sustainability in agroecological studies. The methodology of this case study thus builds on two aspects: first, a quantitative comparison of KE's farming system over time to illuminate changes in land-use and livestock (see Figures 1–3); and secondly, a more qualitative comparison regarding energetic indicators. Together, these two aspects contribute to the overall goal of determining appropriate goals for sustainable management of the agroecosystem from both socioeconomic and agroecological points of view, as proposed by Bromberg, Bertness [44], and Díez, Cussó [6].

In the first part, to analyse the contents of the field books and the livestock (herd lists) and agricultural energy inputs and outputs, all historical manuscripts from the archive on these issues were reviewed and included in an Excel sheet to provide an overview of the content. These data have been entered in a database and georeferenced using ArcGIS Pro [45]. Data and maps available from 1865, 1873, and 1897 [46–48] were digitized in ArcGIS together with a recent topographic map from 2019 [49] and included in an Excel sheet. The data processing is accompanied by interviews with local experts [37,50,51] to verify our interpretation of both the historical and contemporary data.

According to Cattaneo, Marull [32], industrial agriculture has favoured maximizing linear flows, suggesting that product flows out of the system (see Figures 4 and 5). To

describe these flows, the systems boundary is important to define with regards to assessing the linearity (input-output) or circularity (recirculating internally) [19,35,36] of agricultural systems. In this study, we consider the agroecosystem from a single farm standpoint at farm-gate. Thus, modelling the KE system, the system boundaries as defined by the funds and flows accounting for the energy transformation and circulation from an agroecological perspective, as described by Díez, Cussó [6]. The main stocks present in the KE system are a crop livestock system based on historical and current data, with subsystems allowing for the presence of internal flows of energy as described by Hercher-Pasteur, Loiseau [20]. The farmer as the manager or steward of the agroecosystem is part of the external input to the farming system, together with inputs in form of fertilizer, fuel, electricity, heating, and fodder for livestock production. Furthermore, as described by Tello, Galán, Cunfer et al. [52], manure as energy (calculations based on Danish numbers, see [53]) recycled in the system, is also part of the internal flows on this Danish farm. Even if many industrialized system studies exclude human labor [20], the human labor is included in this study, both to enquire about the difference between the KE 1840 and KE 2019 system, but also because, without human labor, there would be no production.

In the second part, the energetic indicators used by Hercher-Pasteur, Loiseau [20], with an anthropic consideration of the system, are regarded as an important qualitative approach to compare sustainability in the system over time. More concrete, the methodology we used follows the flow-fund approach, describing the relationships between elements contributing to the agricultural system as proposed and developed by Cattaneo, Marull [32], including the adoption of the social metabolism concept [32,54]. In the energetic approach of this study, farmland, livestock, farmers, and machinery are regarded as funds, as presented in Table 1.

**Table 1.** Energetic indicators used in this study are cited.

| Formular | References |
| --- | --- |
| $\text{Non-renewable EROI} = \frac{\text{Final Productivity(MJ)}}{\text{Non-renewable external input(MJ)}}$ | [6,18,50,51,55,56] |
| $\text{Final EROI on Labor} = \frac{\text{Final Productivity(MJ)}}{\text{Human labor(MJ)}}$ | [6,57] |
| $\text{External Final EROI} = \frac{\text{Final Productivity(MJ)}}{\text{External Input(MJ)}}$ | [32,35,58] |

As these stocks funds from crop and livestock production can provide flows in form of food, feed, fibre, fuel, and finance, they are capable of providing flows at a given rate, as originally described by Georgescu-Roegen [59] and developed by Cattaneo, Marull [32], and Díez, Cussó [6]. Flows correspond to their respective funds as they need maintenance to be optimized, but also as an overexploitation of flows, which has a negative impact on a systems sustainability. Therefore, as an important energetic indicator, Energy Return on Energy Investment (EROI) (see formula in Table 1 above) from an agroecological point of view is used to examine the energy return relative to the energy used to derive this return in form of energy end products [55–58,60]. As stated by Guzmán, González de Molina [30], from an agroecological point of view, all the biomass produced from a specific agroecosystem should be integrated in the EROI approach. We are aware of this concern, but do not include calculations on biomass in this study, mainly because the historical data does not contain information on biomass, i.e., to avoid that, all calculations would be based solely on estimates, and we therefore choose to omit biomass data. Regarding the energy efficiency and sustainability of agroecosystems, the theoretical concepts described by Tello, Galán [50], Tello, Galán, Sacristán et al. [61], Parcerisas, Dupras [62], and Díez, Cussó [6], are adopted in this study.

## 3. Results

### 3.1. KE in a Comparative View

The results show that KE and Danish agriculture have undergone some significant changes over time. In this results section, we focus the attention, first to land use changes, secondly to the changes in energy flows over time.

In the 1840s, a total of 23% of the land use was pastureland (see Figure 1a,b). In that period, more land was used for cattle or sheep grazing, corresponding to the average national Danish level described by Frederiksen, Rømer, Münier [63]. Field size development shows an increase in average field size from 1.4 ha in 1865 to 5.5 ha in 1873 and further increasing to 10 ha in 1897. This development is mainly due to the incorporation of fallow land into arable land, as confirmed from the maps [47,48]. The development from 1873 to 1897 is regarded as a further development of field drainage and a reclamation of natural areas into arable land.

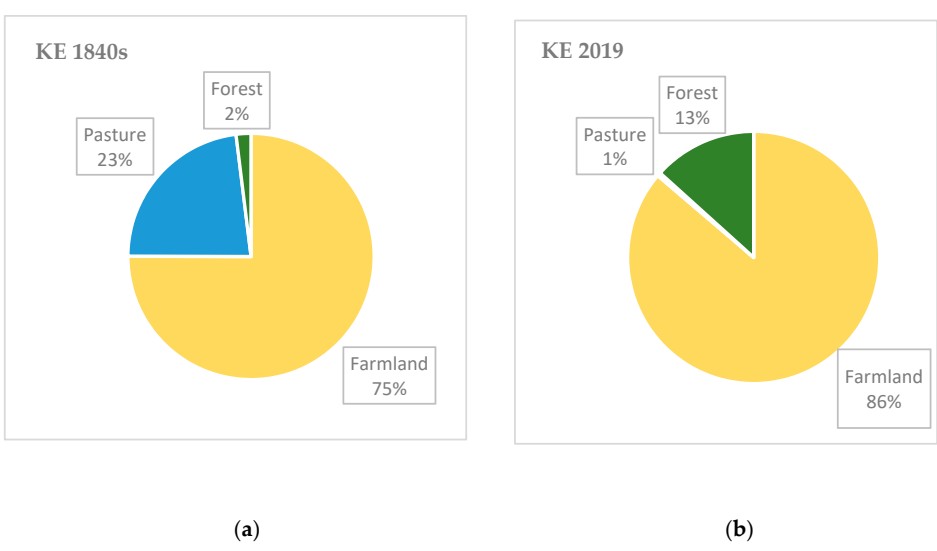

(**a**)　　　　　　　　　　　　　　　　　　(**b**)

**Figure 1.** The distribution of different land-use types accounted for the total area: (**a**) KE 1840s; (**b**) KE 2019.

Hence, especially for meadows, fallow land, and pastures, a sharp intensification of the landscape is observed over time, where the relative percentage of arable area has increased at the expense of the other land-use types, leading to an average field size of 28 ha in 2019. The farm livestock presents a shift from cows and oxen production in the 1840s, to a highly specialized swine production in 2019. The main changes in the farming structure are presented in Table 2 as an overview of the farming community, land-use and livestock data. The farming community decreased significantly over time as peasants are no longer are part of the farm. In 2019, farm-associated people were mainly the operation manager and seasonal workers contributing to sowing at harvest time. Natural areas were ploughed or planted with forest through agricultural intensification throughout the nineteenth century [64]. This is confirmed by the percentage decrease of boundaries between arable land and natural areas from 38% in 1873 to 6% in 2019. At the same time, these results still confirm a very small amount of drainage in arable land at KE in 1873 compared to the whole agricultural area of KE. The increased percentage of field boundaries between arable land and forest has increased from 0% in 1873 to 28% in 2019, corresponding to a higher extent of forest area at KE. Large differences observed in area types can give an indication of change in land use types, and thus in biodiversity associated with natural areas. This development in field size and different land-use types confirms the structural and societal changes towards a continued intensification of industrialized agriculture. The increase in field size, fewer natural areas and fewer field roads are trademarks that confirm this development [58].

**Table 2.** The changing structure of Kragerup Estate's funds 1840s and 2019.

| KE Funds | 1840s | | 2019 | |
|---|---|---|---|---|
| **Farming Community** | | | | |
| Workers | 232 | | 18 | |
| Annual working hours/person | 2570 | | 1983 | |
| Farmland area (ha) (% of total) | 1572 | 75% | 1205 | 86.3% |
| Pastureland (ha) (% of total) | 482 | 23% | 6 | 0.4% |
| Average field size (ha) | 1.4 | | 28 | |
| Forest (ha) (% of total) | 41 | 2% | 186 | 13.3% |
| Total area (ha) | 2095 | | 1397 | |
| **Livestock (Heads)** | | | | |
| Draft animals (Horses) | 18 | | 0 | |
| Cows & oxen | 316 | | 15 | |
| Swine | 20 | | 25,600 | |
| Sheep & lamb | 47 | | 0 | |
| Geese and Hens | 153 | | 0 | |

Even though the KE 2019 system is distinguished by larger field sizes, crop diversity has actually increased over time. In summary, six crop types were present in the field books from the 1840s. In these field books, wheat (Triticum aestivum), barley (Hordeum vulgare), rye (Secale), oats (Avena sativa), and potatoes (Solanum tuberosum) and peas (Pisum sativum) were found. While oats were mainly used as fodder for horses, barley was used both for cooking at KE and to some extent for malting barley. Comparing the 1804s to 2019, the crop variety has expanded (see Figure 2). Besides barley, oats and wheat cultivation now also includes fava beans (Vicia faba), rapeseed (Brassica napus), maize (Zea mays), and a small amount of Christmas trees (mentioned as 'others' in Figure 2) in summary presenting more than seven crop types. The larger number of crops means that the share of each crop has fallen relative to the total area. Rye is no longer grown at all, and the 4% of oats in 2019 is exclusively grown for human nutrition and no longer for horse feed. While, on average, barley was grown on 35% of the total area in the 1840s, the percentage of barley has decreased to 17% in 2019. The average of winter wheat has increased from 12% in the 1840s to 26% in 2019. While the purpose of grass in the 1840s was as a grazing area for the livestock, the purpose in 2019 included grass seed production but not as grazing area for livestock.

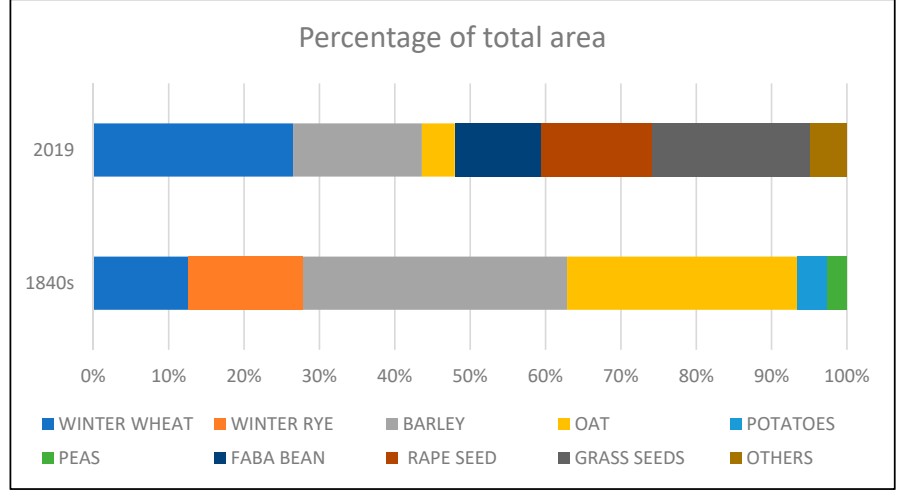

**Figure 2.** Percentage of total area used for different crops 1840s (data from 1841, 1843) and 2019 [39,42].

Crop yield at KE has increased significantly over time. Calculated in absolute numbers, the yield (ton) per ha has increased from the 1840s to 2019; winter wheat from 1.6 to 9.3 ton/ha, barley from 1.3 to 8.1 ton/ha and oat from 1.0 to 7.5 ton/ha. Fields located close to the farm had in the 1840s higher levels of manure fertilization and were utilized more intensively [65].

In the 1840s system, a five to six-fold yield return on grains sown for this specific geographic area was estimated by Dinesen [40] and corresponds to the numbers given for crop yield from Rømer [65], referring only to intensively cultivated crops such as barley, wheat, oats, and rye. The geographical location of the farmland relatively close to the farm also influences the intensity of cultivation (indicated by larger field size close to the manor house). These results can, to some extent, indicate good soil conditions for KE. However, the results also indicate that the soil quality has not only been decisive for which crops the estate and its production focused on. Comparing yield for wheat, barley and oats as the three crops present over time, the average yield increased corresponding to more than a quadrupling. To clarify how this quadrupling of crop yield was obtained, these results were used further on in the calculation on energetic indicators.

KE's historical livestock lists provide information on the number of dairy cows, bulls and oxen registered at KE in the years 1839 and 1844 [66,67]. In the KE farm system, sheep were mainly present at peasant farms as confirmed by the land-rent [68]. When the 1840s are compared to 2019, a significant change in the livestock composition becomes clear as 2019 is distinguished by a focus on specialized pig production (see Figure 3). The amount of cows and oxen decreased significantly over time, while pig production was the main focus of production in the 2019 system.

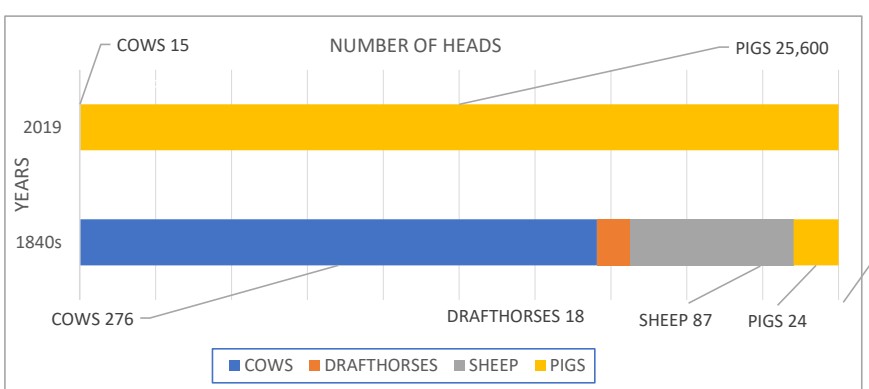

**Figure 3.** KE: total livestock composition according to number of heads [42,47,69].

### 3.2. Energy Funds and Flows over Time

Comparing the 1840s farming system to the 2019 farming system, the transition from pre-industrial to industrialized farming system becomes clear. To give an overview on energy flows in the agroecosystem of the 1840s and 2019, elements from the external input, livestock, and the final produce are listed in Table 3.

The results in Table 3 clearly show the changes with regards to human labor input. These decreased by 92% working hours per hectare and substituting human labor and draft power by modern machinery using non-renewable fuel input. Fertilization supported the significant increase of crop yield per hectare, corresponding to an increase of 407% in the KE farm system when compared over time. Thus, the result was a quadrupled crop yield production at the cost of a high amount of external inputs and potentially environmental consequences of excess nutrient discharge. Livestock production was intensified over time by moving the production from pasture livestock to specialized and industrialized production sites as shown by the decrease of animal diversity and by the increase in pig production.

**Table 3.** Energy flows in the agroecosystem of KE 1840s and KE 2019 in Gigajoule (GJ) and hours (h).

| External Inputs (EI) | 1840s | 2019 |
|---|---|---|
| Fodder | 359 GJ | 12,851 GJ |
| Fuel (Non-Renewable External Input) | 0 | 3629 GJ |
| Electricity & Heating | 0 | 1198 GJ |
| Fertilizer | 0 | 11,906 GJ |
| Human Labor | 455 GJ | 64 GJ |
| Human Labor (hours) | 596,240 h | 35,703 h |
| Sum External Input | 359 GJ | 29,586 GJ |
| Livestock Services | | |
| Draft Power | 147 GJ | 0 |
| Manure | 399 GJ | 3454 GJ |
| Sum Livestock Services | 546 GJ | 3454 GJ |
| Final Produce (FP) | | |
| Crop Production | 29,430 GJ | 114,936 GJ |
| Livestock Production (Meat & Milk) | 895 GJ | 16,194 GJ |
| Sum Final Produce | 30,724 GJ | 134,583 GJ |

From these structural, quantitative data, it is now possible to have a look at possible changes in the overall energetic input and output rates in the farming system (see Table 4).

**Table 4.** EROIs per ha 1840s compared to 2019.

| EROIs | 1840s | 2019 |
|---|---|---|
| External Final EROI (EFEROI) | 38 MJ | 5 MJ |
| EROI on Labor (Labor EROI) | 67 MJ | 2095 MJ |
| Non-Renewable EROI | NA | 58,290 MJ |

Table Note: NA = Not Applicable.

Taking the difference in the size of the farm between the 1840s and 2019 into account, the FP, EI, and labor was calculated per hectare to calculate the relative difference for EFEROI and labor EROI over time. The results show the key importance of labor as being the main force in managing the agroecosystem. Industrialization of livestock production and mechanization of agriculture increased the 1840s labor EROI productivity more than 31 times compared to the Labor EROI in 2019. This increased productivity did, however, not occur along an increased efficiency as shown by the results from the EFEROI. By the significant multiplied increase of EI in the 2019 system, the EFEROI decreased with more than 80%. As the 2019 results still show a number above one, KE has not become a net consumer of energy, but the results do indicate a decreasing farming efficiency for the industrialized agricultural system at KE in 2019.

The relationship between FP and non-renewable external energy input allows one to determine the efficiency of non-renewable energy. The results presented for 2019 show a relatively high efficiency compared to data from the case study from Alonso, Guzmán [53], including greenhouse vegetables, which may explain the lower efficiency. As mentioned by Tello, Galán [60], non-renewable EROI emphasizes the farming systems' dependence on external inputs in relation to the use of non-renewable energy. No non-renewable external inputs were present in the 1840s pre-industrialised system. The results from 2019 show an input of 48 MJ per ha non-renewable energy, which is much higher than the results presented by Pérez Neira [55].

### 3.3. Farm Linearity and Circularity

Figures 4 and 5 show the attributes of the KE agroecosystem's multi-functionality with regards to circularity or linearity for the 1840s and 2019. The farmland, livestock, and farming community provide the basic structure and funds of an agroecosystem. From these funds, the different flows of energy carriers are distinguished and described. In the 1840s farm system, only human labor is present as external input. From the percentage of the total amount of energy output, crop yield represents 96%, livestock represents 4%. From here, the system shows a circularity. Circularity is present by the amount of energy from crop yield or livestock which is returned to the farm. Linear outputs, on the other hand, present the amount of energy leaving the farm as, in this case, sold products in a linear output. The 1840s system shows a high percentage of circularity where crops were re-used as feed for human, fodder for livestock, and seeding for next year's crop production. Also, milk, meat, and manure from livestock production were highly circulated on the farm. The relatively minor amount of linear outputs is represented by mainly barley sold for beer production and studs from the livestock production sold for export.

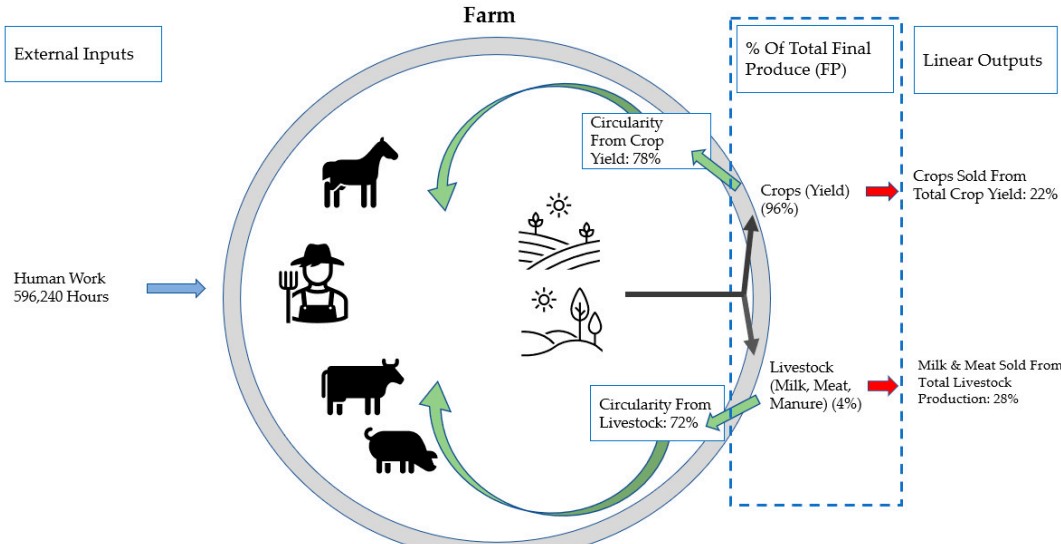

**Figure 4.** Attributes of flows of the KE Agro-Ecosystem 1840s: External inputs to the farm distributed into Final Produce (FP) and from there into circularity or linear output respectively.

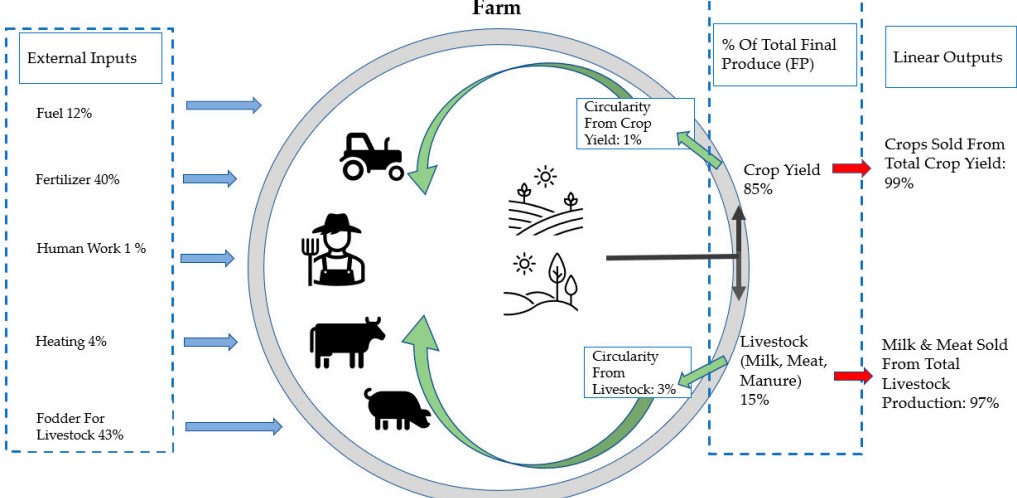

**Figure 5.** Attributes of flows of the KE Agro-Ecosystem 2019: External inputs to the farm distributed into Final Produce (FP) and from there into circularity or linear output respectively.

In 2019, fuel (non-renewable input), fertilizer, heating, and fodder bought from external, local farmers, and necessary for the current livestock production, are present as external inputs to the farm system. The amount of human working hours was 35,703, corresponding to a 94% decrease compared to the 1840s. Human input is, to a much higher degree, represented by management activities and, as drivers of machinery, using fossil fuel in present time much more than in the 1840s. From the percentage of the total amount of energy output, crop yield represents 85% and livestock 15%. From here, the 2019 system shows a high percentage of linearity where 99% from the crop yield was sold from the farm based on fixed contracts. Also 97% of livestock production was sold off the farm.

## 4. Discussion

In the KE case, both appropriation and structural changes are identified, which may argue for several transformations of land-use change being present and shaping modernization. The appropriation of human labor through modern machinery shows that modern, intensive agroecosystems are strikingly different in terms of their structure and function from systems in the 1840s. According to Boserup [69], an increase in the intensity of cultivation is not likely to take place in pre-industrial societies unless there is a population pressure, thus assuming that there are no incentives for farmers to intensify production unless there are tangible benefits to gain from higher yields. Specific factors lead to an intensification of a given system [19]; more concrete, the expansion of arable land is stimulated by economic trends, technologies and demand for cereals. Thus, the division of landscapes into very intensive and extensive agricultural land use types has become more pronounced [62,65].

These changes in land use types is also confirmed by the amount and type of field boundaries, illustrating a change in structural diversity [70] through the establishment of open ditches facilitating the incorporation of fallow land into arable land and the concentration of livestock in stables [71,72]. The decreased proportion of field boundaries towards natural areas in the 2019 system indicates a decrease in structural diversity, which can be expected to have an influence on the overall biodiversity of the system. Even though these structural changes resulted in large, homogenized fields, the KE 2019 farming system includes 14 different crop types compared to six different crop types in 1840, thus presenting an increase in crop diversity in a KE 2019 system well integrated into external markets. An increased variety of crop species correlates to higher diversity [19] and sustainability. Increased species richness thus enhances diversity on the one hand and adds value to the system through resilience, as emphasized by Milestad, Dedieu, Darnhofer, Bellon [73]. Nijs, Impens [74], on the other hand, argue that diversity-productivity relationships and species richness can either increase or decrease productivity through differences in resource utilisation efficiency across species. Here, crop biomass may decrease in systems with mixed species compared to monoculture systems. This would mean that higher diversity does not necessarily benefits all aspects of a farming system. Moreover, a distinction must be made between a structural diversity of natural areas or arable land both regarding field size and the geographical distance between habitats of species. Increased genetic homogeneity increases the potential impact of a variable environment and can increase the damage caused by insects and pathogens as an effect of loss of diversity [75,76]. On the other hand, cultivation of genetically homogenous crops and livestock enhances farming systems productivity and enables the standardization and synchronization of cultural practices in large productions as desirable goals for most industrialized and conventional farms, regarding farm management and marketing [5] and confirming that in the industrial approach, homogenization too often results in the elimination of beneficial relationships and interferences in agricultural systems [19].

KE's land use and livestock contributed to the social and economic development of the local area, whereas industrialisation, synthetic fertilizers, and technological innovation increased productivity and decreased the contribution of human labor input over time, thus changing the impact of the system on local development. The 1840s system mainly

was a local, circular system in the way that products were produced and consumed at KE, the population associated with KE was dependent on KE's agricultural production-related development. In the 2019 system, the local population has been decoupled from agricultural production, as shows the reduction in human labor input reduced to mainly management of the agricultural system. It is debatable whether the decoupling between producer and consumer thus is a positive development, as the decoupling makes the consumer independent from any fluctuations in local production and the associated economy and thus quality of life, or if the social and ecological vulnerability has the greater impact [77].

As shown in the results, adding artificial fertilizer and the use of modern machinery enhances the FP, but on the other hand the dependence of production of non-renewable, costly fossil fuels and the concentrated food production in the hands of few farms challenges sustainable long-term productivity. On this background, sustainability in future agricultural production may become even more challenged through worldwide climate changes, shifts in biosphere and crisis in food-chain supply [78].

The FP is low in the pre-industrial system due to a high input from human labor and no external inputs of fertilizer or modern machinery. Peasantry, representing the largest part of human labor input, scarcely benefited from a surplus. In the KE 2019 system, the FP has increased, mainly due to huge external inputs. Bayliss-Smith (1982) states that the most 'rational' choice for a person will almost always turn out to be the most 'economic,' which could explain the high amount of sold products making the 2019 system to a very linear farming system with import fodder for animal production and seed for crop production from external inputs instead of using crops from own production, as present in the 1840s system. Th industrialization of agriculture thus also increased reliance on external inputs, energy-intensive agriculture [79] and intensification. The result is a decrease in productivity obtained by human management activity [80] at the individual farm level [81]. This development was almost without exception accompanied by growth in farm size and what we here calculated as FP and a decline in human labor requirements [6,74,82,83].

The economic value of a farming area is crucial in order to understand how a landscape heritage is put to play and what a landscape is used for [84]. Changes in operations that lead to successful economic business through increased yield are supported by fixed purchase contracts, as in the KE 2019 case, which are important drivers of the intensification of modern agriculture. Hence, this quantitative intensification may not necessarily coincide with a qualitative as shown by the energetic indicators. This would argue for trade-offs between quality and quantity and elements influencing the system as mentioned in the balance among human nutrition, ecological integrity, and economic development [85]. Hence, the main goals of present-day agriculture seem to be the maximization of production and quantitative maximisation of profit resulting in economic profit through increased productivity. Here productivity challenges some of the 'hard science' metrics for measuring success strictly from productivity gains in form of, for example, higher yields. As mentioned by Woodhouse [13], the agricultural dependency on cheap energy derived from fossil fuel generates a set of questions about the social and environmental sustainability of industrial agriculture. Thus, it can be discussed, if this maximisation also is valid in qualitative terms, if production is not sustainable in its whole, uncertainty regarding non-renewable energy inputs and long-term availability, climate changes, unforeseen forces possibly causing ecological instability and social insecurity. Hence, it can be debated whether this trend towards greater and greater productivity is the desired agroecological future, or whether the focus should be shifting to a greater need and desire for increased circularity, diversity, environmental sustainability, nature and environmental protection and locally produced food.

As the results show, the relationship between productivity and efficiency is not a one-to-one relationship. Modern high input/output agriculture can been questioned in terms of economic costs and impacts on climate [13] and has also be questioned in terms of energy efficiency [6,7,30,86]. It is worth questioning what measurements are needed to increase the efficiency in a way to be effective without compromising productivity, or which compro-

mise would be acceptable. Through this investigation it became clear that cultural energy derived by human labor and draft animals as input alter the energy efficiency, whereas input in form of industrial cultural energy (non-biological) from modern machinery lowers the energy efficiency of the system. Modern machinery is more time-effective but also uses a higher input of energy decreasing the overall EFEROI. In current time, human labor in modern agriculture is justified to be the management part of the system. The answer if a system should be considered from an economic efficiency or from an energy efficiency perspective may vary depending on the viewer's point of view and focus, as already argued by Wilson [7] and Goodman, Sorj [4]. Thus, a schism between efficiency and productivity becomes apparent. This observation must lead to questions about what steps could be taken to increase the energy efficiency while seeking to reduce possible negative impacts on agroecosystems and related socioeconomic parameters. Regarding energy efficiency, it can be discussed if it would be possible to recreate (to some extent) the energy efficiency of pre-industrial systems with a combination of fewer fossil fuel and more renewable energy, which currently is economically viable in society. Kitchen, Marsden [87] argue that aspects of ecological economies and ecological modernization need to be considered also, and they argue for the need to integrate these new elements into the agroecological framework. As stated by Ackoff [88] and supported by Darnhofer, Gibbon, Dedieu [89], parts of systems, considered separately, are made to operate as efficiently as possible, though the system as a whole might not operate as effectively as possible. Thus, the performance of a system depends not on parts acting independently, but on how parts interact with each other.

## 5. Conclusions

The KE farming system has undergone significant changes during the investigated time period and most of the contributing elements have been transformed. Elements of the production process have been appropriated by industrialization. The overall area extent at KE has decreased and technical innovations has enhanced the homogenization of the KE area and herby increased intensity. Inputs from agricultural land use, livestock, and energy systems have all changed radically at KE and changed the system from a highly mixed, circular, and multifunctional pre-industrial system to an industrial, specialised modern farming system. Overall, the KE faming system changed from being a highly circular system to a clear linear farming system. Human labor input decreased by 94% over time, while the external inputs in from of non-renewable energy increased, making the farming system highly productive but less efficient at the same time. As agriculture has developed over time and target functions for the KE farming system have changed over time, the agricultural system of KE today bears little resemblance with the KE system in the 1840s and is challenged with regards to sustainability. Overall, systems are complex arrangements of elements dealing with complex real-life challenges and further investigations, including several cases and aspects are necessary. How agricultural farming developed over time at system level, saying how different agricultural systems influenced sustainability parameters, and how these findings can contribute in a more general form to enlighten the changes in Danish farming systems over time and how these changes at system level can contribute to find sustainable solutions for farming systems in the context of recent environmental and ecological challenges must therefore be seen as a future aim of study.

**Author Contributions:** Conceptualization, (N.L., M.G., C.K.); Methodology (N.L., C.K.); Formal analysis (N.L.); Data curation (N.L.); Writing—original draft preparation (N.L.); writing—reviewing and editing (N.L., M.G., C.K.); Supervision (M.G., C.K.). All authors have read and agreed to the published version of the manuscript.

**Funding:** This study was supported by the ProvenanceDK Project with funding from Innovation Fund Denmark (grant number 6150-00035B) and the SustainScapes project, https://bio.au.dk/forskning/forskningscentre/center-for-sustainable-landscapes-under-global-change/ (accessed on 15 November 2021) funded by the Novo Nordisk Foundation through their "Novo Nordisk Foundation Challenge Programme" project No. 35840 SUSTAINSCAPES.

**Institutional Review Board Statement:** Not applicable.

**Informed Consent Statement:** Not applicable.

**Data Availability Statement:** The data presented in this study are available on request from the corresponding author. The data are not publicly available due to private ownership.

**Acknowledgments:** We gratefully thank Tommy Dalgaard, Aarhus University, Niels Mark Jacobsen, Aarhus University and Huayang Zhen, China Agricultural University for assistance and useful suggestions. Many thanks also to our anonymous reviewers for the invaluable constructive criticism and suggestions.

**Conflicts of Interest:** The authors declare no conflict of interest.

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
