# Peer review of "Historical Transition of a Farming System towards Industrialization: A Danish Agricultural Case Study Comparing Sustainability in the 1840s and 2019"

_sustainability, doi:10.3390/su132212926_

Round 1
Reviewer 1 Report
The manuscript 'Historical transition of a farming system towards industrialization: A Danish agricultural case study comparing sustainability in the 1840s and 2019' is showing light on the agricultural development of Danish farming system. Please consider the following points and revise the manuscript accordingly.
Abstract: Rewrite the abstract section with more study findings in terms of facts and figures.
Keywords: remove keyword - linearity and circularity of a Danish farming
system and mention as Danish farming system
Introduction:
- Add more review of the literature and their results
- Introduction part is missing rationality of the study
- Add a sequence of agricultural development in the Danish agricultural system
Methodology
- Provide detailed information on Kragerup Estate like lat long information, type of farming system, major crop and livestock, type of nutrient management, irrigation management, soil properties (if available) over a period of time in a tabular form
- Definition in the methodology part is redundant remove from line 141-143
Results
- Very well written with sufficient figures
- Do you need to cite the past publications in the results section. I wont think it is required. Please take a decision on this with appropriate reasoning
Discussion
- The discussion part is too lengthy, reduce it suit the results
Reviewer 2 Report
The manuscript is significant in that it excavates a historical case to verify the known changes between traditional and conventional agricultural patterns. but it quite provides just a comparison with record data, the deep background and implication of the Danish agricultural evolution should be addressed further, as well as what do the findings mean to the future Danish agriculture under the school of agroecology discipline, and the discussion section could be more succinct. Also, there is a word "drafthorses" missed in Fig.3.
Author Response
SECTION |
NO. |
REVIEW COMMENT |
RESPONSE FROM AUTHOR |
LOCATION OF CHANGES (line) |
General |
1 |
Please add a section named "Author Contributions" after the "Acknowledgments" section in your paper. |
Section added |
528-530 |
2 |
Please state the individual contribution of each co-author to the reported research and writing of the paper. e.g., who designed research; who performed research and analyzed the data; who wrote the paper. All authors read and approved the final manuscript. |
stated |
527-529 |
|
Reviewer 1 |
|
|
|
|
Abstract |
3 |
Rewrite the abstract section with more study findings in terms of facts and figures. |
Results from findings are added to the abstract.
|
15-19 |
Introduction
|
4 |
Add more review of the literature and their results
|
added |
31-45 |
5 |
Introduction part is missing rationality of the study
|
See changes for comment 4 |
31-45 |
|
6 |
Add a sequence of agricultural development in the Danish agricultural system
|
added |
46-51 |
|
Keywords |
7 |
remove keyword - linearity and circularity of a Danish farming |
Keyword is removed. |
23 |
Methodology
|
8 |
Provide detailed information on Kragerup Estate like lat long information, type of farming system, major crop and livestock, type of nutrient management, irrigation management, soil properties (if available) over a period of time in a tabular form |
Described with a short text |
121-127 |
9 |
Definition in the methodology part is redundant remove from line 141-143 |
Lines are removed. |
|
|
Results
|
10 |
Very well written with sufficient figures |
Thank you. |
|
11 |
Do you need to cite the past publications in the results section. I won’t think it is required. Please take a decision on this with appropriate reasoning |
|
Repeated references were removed, but references necessary for understanding of content were kept in the text |
|
Discussion |
12 |
The discussion part is too lengthy, reduce it suit the results |
Discussion is reduced |
|
Reviewer 2 |
|
|
|
|
|
13 |
The deep background and implication of the Danish agricultural evolution should be addressed further
|
A section is added in the introduction |
47-51 |
14 |
as well as what do the findings mean to the future Danish agriculture under the school of agroecology discipline |
perspectivation added at the end of conclusion section |
512-515 |
|
15 |
and the discussion section could be more succinct |
See note to comment 12 |
|
|
16 |
there is a word "drafthorses" missed in Fig.3 |
added |
Fig 3 |

Round 2
Reviewer 1 Report
There is still scope to improve the discussion part, shorten the discussion part.
Author Response
Dear reviewer.
The discussion part is worked through, scoped to improve and shortened especially in the first part (see line 356 - 385). Some more improvements are done in line 403, 417, 433 -438. Please see all the track changes in the attachment.
Best,
Nele
